# A Systematic Review of Complementary and Alternative Veterinary Medicine: “Miscellaneous Therapies”

**DOI:** 10.3390/ani11123356

**Published:** 2021-11-24

**Authors:** Anna Bergh, Iréne Lund, Anna Boström, Heli Hyytiäinen, Kjell Asplund

**Affiliations:** 1Department of Clinical Sciences, Swedish University of Agricultural Sciences, SE 750 07 Uppsala, Sweden; 2Department of Physiology and Pharmacolgy, Karolinska Institutet, SE 171 77 Stockholm, Sweden; irene.lund@ki.se; 3Department of Equine and Small Animal Medicine, Faculty of Veterinary Medicine, University of Helsinki, P.O. Box 57, 00014 Helsinki, Finland; anna.bostrom@helsinki.fi (A.B.); heli.hyytiainen@helsinki.fi (H.H.); 4Department of Public Health and Clinical Medicine, Umeå University, SE 901 87 Umeå, Sweden; kjellasplund1@gmail.com

**Keywords:** aromatherapy, gold therapy, homeopathy, leeches, mesotherapy, mud, neural therapy, music therapy, vibration therapy

## Abstract

**Simple Summary:**

Complementary and alternative veterinary medicine (CAVM) is commonly used in animals. However, there is limited knowledge of how the methods affect the animal. Therefore, this study reviews the scientific literature of 24 different CAVM therapies used in cats, dogs, and horses. Three core bibliographic sources were used. Relevant articles were assessed for scientific quality, and information was extracted on study characteristics, species, type of treatment, indication, and treatment effects. Of 982 unique publications screened, 42 were eligible for inclusion, representing nine different CAVM therapies, which were aromatherapy, gold therapy, homeopathy, leeches (hirudotherapy), mesotherapy, mud, neural therapy, sound (music) therapy, and vibration therapy. For the other 15 predefined therapies, no study was identified. The risk of bias was assessed as high in 17 studies, moderate to high in 10, moderate in 10, low to moderate in four, and low in one study. In those studies where the risk of bias was low to moderate, there was considerable heterogeneity in reported treatment effects. Therefore, the 24 CAVM therapies do not have scientific documentation of sufficient quality to draw clear conclusions regarding their clinical effect.

**Abstract:**

There is an increasing interest in complementary and alternative veterinary medicine (CAVM). There is, however, an uncertainty of the efficacy of these methods. Therefore, the aim of this systematic literature review is to assess the evidence for clinical efficacy of 24 CAVM therapies used in cats, dogs, and horses. A bibliographic search, restricted to studies in cats, dogs, and horses, was performed on Web of Science Core Collection, CABI, and PubMed. Relevant articles were assessed for scientific quality, and information was extracted on study characteristics, species, type of treatment, indication, and treatment effects. Of 982 unique publications screened, 42 were eligible for inclusion, representing nine different CAVM therapies, which were aromatherapy, gold therapy, homeopathy, leeches (hirudotherapy), mesotherapy, mud, neural therapy, sound (music) therapy, and vibration therapy. For 15 predefined therapies, no study was identified. The risk of bias was assessed as high in 17 studies, moderate to high in 10, moderate in 10, low to moderate in four, and low in one study. In those studies where the risk of bias was low to moderate, there was considerable heterogeneity in reported treatment effects. Therefore, the scientific evidence is not strong enough to define the clinical efficacy of the 24 CAVM therapies.

## 1. Introduction

Complementary and alternative veterinary medicine (CAVM) is a term describing a range of therapies that vary in theory and practice. It comprises therapies that are mainly delivered by therapists without a veterinary medical background, and to a minor degree or not at all by animal health personnel. In human medicine, the term CAM is used in contrast to “conventional medicine”, defined as scientifically evidence-based medicine or “well-documented experience”. A key difference compared to conventional medicine is that CAVM methods use explanatory models for their mechanisms of action and clinical effect that either differ from those of conventional medicine or are disputed. An example can be the explanation model with the balancing of energies used in traditional Chinese medicine compared with the activation of endogenous systems with the release of e.g., different neuropeptides used in western medical acupuncture. Based on human literature, there are certain therapies that do not have enough scientific support to determine clinical efficacy but whose mechanisms of action can be explained by natural sciences. These therapies might be more relevant to explore scientifically than those that rest solely upon an explanatory model that is not compatible with existing scientific models. Further, there are also certain therapies for which there is some evidence regarding clinical efficacy in humans, but there is a lack of evidence when it comes to the treatment of animals.

The documentation is also limited regarding how commonly CAVM therapies are used. Some studies report that interest in CAVM has increased in recent years [1,2,3,4]. A study conducted in New Zealand investigated the use of and attitudes towards allied health therapy via interviews with 110 horseback riders and trainers. It showed that 62% used physiotherapy, chiropractic, or massage on their horses. The most commonly used treatment was chiropractic, and the most common indication was equine back pain. Only 7% of respondents chose the relevant method on the recommendation of their veterinarian, and as many as 72% stated that the veterinarian and the therapist did not cooperate during the treatment [2]. In addition, Bergenstrahle and Nielsen (2016) report that a minority of equine veterinarians, answering a survey, had taken CAVM classes during college [5]. Further, the supply of services offered at veterinary clinics has historically been driven by demand from animal owners [6]. Due to the demand from animal owners, veterinarians and other animal health personnel should be well prepared for questions regarding CAVM. There is therefore a need for unbiased information about the possible effects and consequences of the use of CAVM. The aim of the present article is to make a systematic literature review of CAVM therapies that are either rare or do not belong to a specific group of therapies (such as electrotherapies) and are therefore called “miscellaneous therapies”. To the best of the authors’ knowledge, only a limited number of previous systematic reviews match the scope of this article. The exceptions are one meta-analysis, one systematic veterinary medicine literature review of RCTs of homeopathy trials, and a systematic literature review of RCTs in gold therapy [7,8,9]. 

The present article is one of a future series of systematic review articles in a special issue of *Animals* on CAVM therapies used in companion animals (cats, dogs, and horses). The other articles will focus on manipulation/mobilization therapies, electrotherapies, soft tissue mobilization, and acupuncture. In the present article, we have collected scientific literature for a number of CAVM interventions, including anthroposophic medicine, aromatherapy, bioresonance, body work, colloidal silver, crystal therapy, healing, gold therapy, healing touch, homeopathy, infrasound, ion therapy, iridology, kinesiology, leeches, mesotherapy, mud therapy, naprapathy, neural therapy, sound therapy, therapeutic touch (a non-contact healing method), vacuum therapy, vibration therapy, reflexology, and zone therapy.

## 2. Materials and Methods

The overall outline of our systematic review adhered to the Cochrane guidelines on how to perform a systematic review [10], as adapted by the Swedish Agency for Health Technology Assessment and Assessment of Social Services (SBU) in its methodological handbook [11]. 

### 2.1. Review Topic/Research Question

To assess the evidence for clinical efficacy of complementary and alternative veterinary medicine (CAVM) therapies used in companion animals, the article covers a range of therapies not included in other reviews in this special issue. 

### 2.2. Search Strategy

Professional librarians performed searches of Web of Science Core Collection, CABI, and PubMed (1980–2020) in August 2020. The keywords were terms relevant to dog OR cat OR horse, AND veterinary medicine OR veterinary, AND therapy* OR treatment*, AND anthroposophic medicine OR aromatherapy* OR bioresonance therapy* OR body work OR colloidal silver OR crystal therapy* OR distance healing OR gold therapy* OR healing touch OR homeopathy OR infrasound therapy* OR iridology OR ion therapy* OR ion therapy* OR kinesiology OR leaches OR leeches OR mesotherapy* OR mud OR naprapathy* OR neural therapy* OR sound/therapy OR sound therapy* OR sound/therapeutic use OR therapeutic touch OR vibration/therapeutic use OR zone therapy* OR reflexology* (* indicates truncation).

### 2.3. General Inclusion and Exclusion Criteria

The inclusion criteria were that the study must be published in a peer-reviewed journal, be accessible by the authors through institutional access or internet search, and be an original research publication. There were no restrictions with regard to either country or language of publication at the initial search stage. The study should describe the efficacy of one of the above-mentioned therapies in the treatment of a single indication in cats, dogs, or horses. The study design could be randomized controlled trials (RCTs), other interventional studies, or observational studies. Laboratory experimental studies were included only if the study mimicked a clinical situation and/or a mechanism of action was studied. Abstracts were included if no full-length article was available. Case studies were included only if five or more subjects were reported. 

The exclusion criterion was any publication that involved more than one type of treatment. 

### 2.4. Study Selection and Categorization

All screening was performed based on journal title, publication title, or abstract. Citations identified were imported into Endnote (X9.3.3, 2018), and duplicates were removed. A single author (AB) applied inclusion and exclusion criteria to all publications. 

In the screening phase, we identified articles of possible relevance for the review; articles describing one type of CAVM intervention in cats, dogs, or horses were selected for full text reading. A therapeutic intervention was defined as an intervention intended to reduce the signs, severity, or duration of a clinical condition. After the first stage of screening, articles deemed potentially relevant were accessed from open-access sources. Articles that could not be accessed from digital library resources were requested via the Swedish University of Agricultural Sciences library. If the full manuscript was not found following these steps, but an abstract was available, then categorization was undertaken based on the abstract. For each study, the following key descriptive items were tabulated using templates modified after SBU [11]: first author, year of publication, study design, study population, intervention, type of control, outcome, and relevance (external validity).

Assessment of the risk of bias (scientific quality) of each article was performed in accordance with the Cochrane [10] and SBU [11] guidelines. The assessment was based on the following items: study design, statistical power, deviation from planned therapy, lost to follow-up, type of outcome assessment, and relevance. In the assessment of observational studies, risk of confounding was also included. The manual used for assessment of risk of bias and relevance is available as a Appendix A. For consistency, before starting the literature review, three of the authors (KA, HH, AB) independently screened a random sample of articles, and differences were discussed and resolved before reviewing all articles.

## 3. Results

A total of 1222 articles were identified via the three combined electronic database e-searches (see Figure 1). 

The search results were as follows: anthroposophic medicine (1 article), aromatherapy (5), bioresonance (1), body work (6), colloidal silver (7), crystal therapy (22), distance healing (4), gold therapy (145), healing touch (3), homeopathy (644), infrasound (10), ion therapy (5), iridology (0), kinesiology (98), leeches (54), mesotherapy (33), mud therapy (10), naprapathy (0), neural therapy (23), sound therapy (69), therapeutic touch (28), vacuum therapy (32), vibration therapy (16), reflexology (0), and zone therapy (6). Following the removal of duplicate articles, 982 articles were screened for relevance to the review. A large proportion of studies were excluded since they were either published in proceedings or textbook chapters, were experimental studies not mimicking a clinical indication, or were case studies with less than five animals. Following the title and abstract screening, 114 publications investigating treatment of a single indication with any of the listed miscellaneous therapies were evaluated in full. After completion of the selection process, 42 articles concerning nine therapies were retained. The data on the included articles is presented in Table 1, Table 2, Table 3, Table 4 and Table 5, with a description of the article´s content and the level of risk of bias. For the following 15 therapies, no articles were found to fulfil the inclusion criteria: anthroposophic medicine, bioresonance, body work, colloidal silver, crystal therapy, distance healing, healing, infrasound, ion therapy, iridology, kinesiology, naprapathy, therapeutic touch, vacuum therapy, and zone therapy.

A total of 42 publications described treatment for 23 different indications, with the largest number of studies regarding treatment of osteoarthritis (6), followed by stress (5), dermatitis (3), and back pain (3). There were 19 publications on horses, 22 on dogs, and four on cats (some including both cats and dogs). In total, the number of animals investigated was 334 horses, 650 dogs, and 63 cats. For horses, the most commonly studied treatment was stress reduction with aromatherapy (3), followed by music therapy (1), different types of lameness treated with vibration therapy (1), and leeches (1). Finally, studies on sound horses investigating treatment effects on different physiological variables such as bone density, movement, and on blood variables were included in the systematic review when other articles were lacking. For dogs, it was treatment of osteoarthritis with gold therapy (4) and homeopathy (2), followed by atopic dermatitis with homeopathy (2) and neural therapy (1), and back pain with mesotherapy (2). For cats, the treatment of malignancy with gold nano-rods illuminated with light was the most commonly studied therapy. 

### 3.1. Aromatherapy

Aromatherapy is a treatment that uses natural plant extracts to promote health and well-being. 

Four RCTs, involving a total of 24 horses and five dogs (but none on cats), were identified (see Table 1). 

#### 3.1.1. Study Quality

In three studies, the risk of bias was assessed as moderate to high, and in one study, it was assessed as high. The high risk of bias was mainly attributed to low numbers of participants, resulting in insufficient statistical power and suboptimal outcome measures.

#### 3.1.2. Clinical Indications

In all four studies, the method was used for stress reduction in otherwise sound horses and dogs. 

#### 3.1.3. Interventions and Controls

The three equine studies involved either aromatherapy with humidified lavender oil or chamomile oil, or just humidified lavender oil, with humidified air as the control. The dosages varied between studies: (1) 100% lavender and chamomile oil, respectively, in a diffuser held under the horse’s nose over two hours; (2) 20% lavender oil, administered by a humidifier for 15 min under the horse’s nose. During this period, an air horn was blown twice, 15 s each time; (3) the same type of lavender dosage (20%), but administered during a 15-min trailer ride. The study on dogs administered lavender oil (0.18 mL) or saline (0.9% NaCl) solution (0.18 mL) topically to the inner pinna of both ears, four times for one day, with no control group. 

#### 3.1.4. Outcome Variables

The efficacy of aromatherapy was assessed by monitoring either heart and/or respiratory rate, or heart rate variability. In one study, cortisol and norepinephrine concentration was added to the heart rate registrations. The canine study used an ambulatory ECG monitor to register heart rate and analyzed spectral indices of heart rate variability, power in the high-frequency range, and the ratio of low-frequency to high-frequency power.

#### 3.1.5. Clinical Effects

In the first equine study (high risk of bias), aromatherapy with humidified lavender oil showed a statistically significant difference in heart rate variability compared to the chamomile oil and control. The second equine study (moderate to high risk of bias) showed a significantly lower heart rate after lavender oil than control. The third (moderate risk of bias) showed no difference in heart rate, but a significantly lower concentration of cortisol compared to the control. The canine study reported inconclusive results (see Table 1).

### 3.2. Gold Therapy

Injectable gold, most commonly used to treat osteoarthritis. 

Seven publications were retrieved, of which four reported on RCTs in a total of 218 dogs. Three case series involved a total of 30 dogs and eight cats. No study on horses was identified (see Table 2). 

#### 3.2.1. Study Quality

Of the four RCTs, two were graded as having low to moderate risk of bias, one moderate and one high. The case series reports were assessed as having high (one study) or moderate to high risk of bias (two studies). The main reason for high risk was low numbers of participants, resulting in insufficient statistical power, as well as lack of control groups.

#### 3.2.2. Clinical Indications

In two case series, dogs and cats with natural mammary gland tumors were treated with gold nanorod-assisted plasmonic photothermal therapy (PPTT), with no control group. In one case series on dogs, gold wires were used to treat epileptic seizures. The remaining four studies, all RCTs, investigated the effect of gold bead or wire implantation for dogs with osteoarthritis (OA). 

#### 3.2.3. Interventions and Controls

In the two case series studies evaluating PPTT, the treatment was (1) one intratumoral injection of 75 μg gold nanorods/kg of body weight followed by direct exposure to 2 W/cm^2^ near infra-red laser light for 10 min on ablation of mammary tumor; or (2) three sessions of PPTT treatment at two-week intervals with a gold concentration of 7.5 nM. Both studies used an 808 nm diode laser. In the case series study on epileptic seizures, gold wire implants at acupuncture points were used, with no control group. The four RCTs used gold bead or wire implantation around the hip joint, with needle “holes” as control. 

#### 3.2.4. Outcome Variables

In the PPTT studies, outcome was assessed via histopathology, diagnostic imaging, laboratory blood analysis, and comprehensive clinical examinations. In the seizure study, outcome was assessed by EEG and a protocol registering the number and intensity of seizures before and after treatment. In the OA studies, clinical examination, owner questionnaires, and kinetic and kinematic analysis were conducted.

#### 3.2.5. Clinical Effects

In the PPTT studies (with moderate-to-high risk of bias), the histopathological results showed a reduction in cancer grade and remission of the tumors. The seizure study (with high risk of bias) showed no changes in EEG recordings but a significant change in owner-reported number of seizures between treatment and control periods. Two of the OA RCTs, graded as being of high and moderate risk of bias, respectively, showed improvements in mobility and reduction of pain for the treated dogs compared to the placebo group. In contrast, the two other RCTs, both with low to moderate risk of bias, showed no differences between the intervention and control groups (see Table 2).

### 3.3. Homeopathy

Two of the core suppositions of homeopathy are “Like cures like”—the notion that a disease can be cured by a substance that produces similar symptoms in healthy animals and people, as well as “Law of minimum dose”—the notion that the lower the dose of the medication, the greater its effectiveness. 

Fifteen publications were retrieved, of which seven reported on RCTs in a total of 213 dogs and 40 cats. Five case series and one case-control study involved a total of 67 horses, 15 cats, and 63 dogs. A prospective observational cohort study involved 68 dogs (see Table 3). 

#### 3.3.1. Study Quality

Of the seven RTCs, one was graded as having low risk of bias, three low to moderate, two moderate to high, and one as having high risk of bias. The case series reports, case-control study, prospective observational cohort study, and retrospective study were assessed as having high risk of bias, except for two studies that were graded as having moderate to high risk of bias. The main reason for the assessment “high risk of bias” was low numbers of participants, resulting in insufficient statistical power. Other factors were lack of controls and a high risk of confounding factors affecting the results.

#### 3.3.2. Clinical Indications

Fifteen publications described use of homeopathy for 13 types of indications. One canine study each investigated the effects on asymptomatic heart failure, pseudopregnancy, fear of fireworks, babesiosis, oral papillomatosis, and problems in the immune system. Two studies evaluated effects on OA and atopic dermatitis, respectively. One feline study each investigated eosinophilic granuloma complex and hyperthyroidism. One equine study each looked at stereotypic behaviour, lameness, and laminitis.

#### 3.3.3. Intervention and Controls

The canine study that investigated the hypotensive effect of homeopathy on early (stage B 2) heart failure treated the dogs with Crataegus oxyacantha at a potency of (1) 6cH and (2) Crataegus MT or (3) hydroalcoholic solution (placebo). The canine study on pseudopregnancy treated Group I with: Thuja occidentalis D30 (eight globules, three times a day, per os); Group II: Urtica urens D6 (eight globules, three times a day, per os); Group III: naloxone (control group, 0.01 mg/kg, twice daily, s.c.). The one on fear of fireworks treated subjects with a potentised homeopathic remedy based on phosphorus, rhododendron, borax, theridion, and chamomilla (6C and 30C in 20% alcohol), and a control (placebo) preparation of water and 20% alcohol in an identical bottle with integrated dropper. The one on babesiosis treated Group A with C. horridus 200C, four pills four times a day orally for 14 days, and Group B with diminazene aceturate at 5 mg/kg intramuscularly single dose. All the dogs were administered 5% dextrose normal saline at 60 mL/kg intravenously for four days. The study on oral papillomatosis used homeopathic remedies in combination (Sulfur 30C, Thuja 30C, Graphites 30C, and Psorinum 30C) and placebo (distilled water), administered orally twice daily for 15 days. The study on problems in the immune system treated subjects with a basal diet with an additional dose of 0.5 mL/animal/day homeopathic solution (Echinacea angustifolia 6 CH, Aconitum napellus 30 CH, Veratrum album 30 CH, Pyrogenium 200 CH, Calcarea carbonica 30 CH, and Ignatia amara 30 CH) and controls received only the basal diet (300 g/day). 

Two canine studies evaluated the effects on OA and atopic dermatitis, respectively. The dogs with OA were treated with the complex homeopathic preparation Zeel ad us.vet (one to three tablets orally per day depending on body weight) and placebo (lactose capsule) and as active control carprofen (4 mg/kg body weight) over 56 days. The dogs with atopic dermatitis were treated with a combination product Adrisin, three times a day over three weeks, and individualised remedies were prescribed based on the dog’s cutaneous signs and constitutional characteristics. 

One feline study investigated the effect on eosinophilic granuloma complex and hyperthyroidism. These included treatment with (1) snake remedies Lachesis (nine cases), Crotalus cascavella (1), Crotalus horridus (1), Cenchris contortrix (1), Elaps corallinus (1), Naja (1) and Vipera (1) for the eosinophilic granuloma complex, and (2) sarcode thyroidinum and an appropriate individualised simillimum using information from a constitutional questionnaire for the hyperthyroidism and a placebo. 

One equine study evaluated the effects on stereotypic behaviour, lameness, and laminitis. The treatments were (1) Ignatia and/or Gelsenium, Stramonium, Phosphorus, Nux vomica, Pulsatilla, Hypericum, Lycopodium, Argentum nitricum, Staphysagria, Arsenicum album, Lachesis, and Thuya occidentalis as treatment remedies specific for each horse; (2) hyaluronic acid (control) and complex of 14 homeopathically-prepared ingredients (Zeel, D8); (3) Aconitum 30C, Apis 15C, Arnica 7C, Belladonna 9C, Bryonia 9C and Nux vomica 9C (two granules of each component were administered 10 times per day for 10 days).

#### 3.3.4. Outcome Variables

The outcome variables for the canine studies were clinical examination, blood analysis, ECG, different types of assessment scales, owner questionnaires, and force plate. The outcome variables for the feline studies were clinical examination and T4 values. The outcome measures for the equine studies were clinical examination, owner questionnaires and blood analysis. 

#### 3.3.5. Clinical Effects

The canine study, of moderate to high risk of bias, which investigated the hypotensive effect of homeopathy on early-stage (B2) myxomatous mitral valve disease showed no difference between groups. The one on pseudopregnancy, concerning mammary gland scores, yielded significantly higher success rates in treated groups compared to the control group (moderate-to-high risk of bias). The one on fear of fireworks showed no difference between groups, based on owner’s rating (moderate to high risk of bias). The one on babesiosis, with high risk of bias, showed no difference between groups. The study on oral papillomatosis showed early recovery with a significant reduction in oral lesions reflected by clinical score in comparison to the placebo-treated group (moderate-to-high risk of bias). The study on problems in the immune system, with high risk of bias, reported that lymphocyte counts were greater in the treatment group on days 30 and 45 of the experiment. The two studies on atopic dermatitis, as well as one on OA, showed no difference between groups (all three studies with high risk of bias). In the remaining low to moderate risk of bias study on OA, when the outcome was measured by dichotomous responses of ‘improved’ or ‘not improved’, three out of the six variables showed a significant difference in improved dogs per group between the treated group and the placebo group. 

One feline study investigated the effect on the eosinophilic granuloma complex (high risk of bias) and hyperthyroidism (low risk of bias), with negative results. 

One equine study on stereotypic behavior showed, by owner reports, a decrease in stereotypic behavior (high risk of bias). The one on lameness showed a reduction in lameness based on clinical examination (moderate to high risk of bias). The one on laminitis, with high risk of bias, showed a clinical improvement after one day (see Table 3). 

### 3.4. Leeches (Hirudotherapy)

Hirudotherapy is a treatment using medicinal leeches. 

One retrospective cohort study involving 57 horses was identified (see Table 4). 

#### 3.4.1. Study Quality

The publication had high risk of bias due to its study design with a lack of control group and a high risk of confounding factors. 

#### 3.4.2. Clinical Indication, Intervention and Control, and Outcome Variables

The study described the treatment with 117 leech applications with many different types of treatment protocols, in 57 horses with laminitis. An Obel scale assessed the efficacy of the treatment.

#### 3.4.3. Clinical Effects

A total of 84% of the horses showed clinical improvement after treatment, based on this high risk of bias study. 

### 3.5. Mesotherapy

Mesotherapy is a non-invasive non-surgical technique that uses micro-injections of pharmaceutical and homeopathic preparations, plant extracts, vitamins, and other ingredients into subcutaneous fat.

One RCT involving 15 dogs and one retrospective cohort study involving 20 dogs were identified (see Table 4).

#### 3.5.1. Study Quality

The RCT study was judged as having a moderate risk of bias and the retrospective study as moderate to high risk, due to the study design and low numbers of participants.

Clinical indications. 

The method was used in dogs to treat chronic back pain.

#### 3.5.2. Interventions and Controls

The RCT study compared active mesotherapy treatment (*n* = 10) with control treatment (*n* = 5). The active treatment group received a combination of 140 mg lidocaine, 15 mg dexamethasone, and 20 mg thiocolchicoside along with a 70-day course of a placebo, administered as if it was carprofen. The control group received carprofen for 70 days, at a dose adjusted to their weight. Further, on day 0, an intradermal injection of Ringer’s lactate was also administered. Animals in both groups rested for three days and resumed normal activity over a five-day period. The retrospective study used (1) a combination of lidocaine, dexamethasone, and thiocolchicoseide, and (2) as previous with an additional traumeel LT.

#### 3.5.3. Outcome Variables

Response to treatment, measured by the Canine Brief Pain Inventory (CBPI) and the Hudson Visual Analogue Scale (HVAS), was evaluated before treatment, after 15 days, and after one, two, three, four, and five months.

#### 3.5.4. Clinical Effects

When comparing CBPI results in the RCT study (with a moderate risk of bias), no differences were found between the treatment group (TG) and control group at baseline through two and five months. Differences were observed in CBPI sections after the discontinuation of carprofen: at three months for Pain Interference Score (PIS) and Pain Severity Score (PSS) and at four months for PIS and for PSS, with group TG having overall better results. No differences were registered with the HVAS. In the retrospective study (with high risk of bias), no differences were observed between groups.

### 3.6. Mud Therapy

Mud therapy involves treating conditions with mineral-rich water mud.

One case study involving 10 sound horses was identified (see Table 4). 

#### 3.6.1. Study Quality

The publication had high risk of bias due to a study design with a lack of control group, limited number of participants and a high risk of confounding factors possibly affecting the result. 

#### 3.6.2. Clinical Indication, Intervention and Control, and Outcome Variables

The method was used in horses to enhance joint flexibility and movement. Horses were treated with mud from Lake Hévíz in Hungary 10 times, twice daily in the evenings. Before and after the experiment and eight weeks following it, measurements were taken of the average stride length and the longest distance between the hind and front foot during walking and trotting, and maximal flexibility of knee, hock, and fetlock joints. The maximal flexibility of each joint was measured with a joint protractor. There were no controls included in the study.

#### 3.6.3. Clinical Effects

The authors of this high risk of bias study reported that the mud treatment had a positive and durable effect on the joints and movement.

### 3.7. Neural Therapy

Neural therapy is a treatment in which local anesthetic is injected into certain locations of the body.

Two case studies involving 18 dogs and 60 horses were identified (see Table 4).

#### 3.7.1. Study Quality

The studies’ risk of bias was judged as high, due to the study design (case studies) and no control groups.

#### 3.7.2. Clinical Indications

Neural therapy (NT) was used for canine atopic dermatitis in 18 dogs and for pain syndrome in the loin and hip region of 60 horses.

#### 3.7.3. Interventions and Controls

The dogs were treated with one set of NT, given by injecting an intravenous dose of 0.1 mg/kg of a 0.7% procaine solution, followed by 10 to 25 intradermal injections of the same solution in a volume of 0.1–0.3 mL per site. Dogs were given between six and 13 sets of injections. There were no controls. The horses were treated with lidocaine, 5 mL of a 1% solution without additives for each point, usually by segmental infiltration of skin, muscles, and spinal nerve roots at eight to 14 segments. This infiltration was repeated each third day, four to five times. There were no controls.

#### 3.7.4. Outcome Variables

In the canine study, the dermatological condition of each patient was evaluated before and after the treatment using two scales: the pruritus visual analogue scale (PVAS) and the canine atopic dermatitis extent and severity index (CADESI). In the equine study, clinical examination was used as an outcome measure, as well as return to racing.

#### 3.7.5. Clinical Effects

In the canine study (high risk of bias), the reduction of pruritus was significant comparing before and after treatment. The equine study (with high risk of bias) reported that of the 60 patients, 51 horses were treated with infiltration, and of those, 45 were assessed after treatment. Seven of them were no longer used for competition, and in four horses, the evaluation time after treatment was too short. Of the remaining 34 horses, 26 could be trained successfully and won several races through the following years, while eight horses did not recover.

### 3.8. Sound (Music) Therapy

Sound (music) therapy uses sound and music to improve health and wellbeing.

One RCT involving 60 sound horses was identified (see Table 4). 

#### 3.8.1. Study Quality

The study´s risk of bias was high due to study design. Further, performance variables such as success coefficient in racing season could have been influenced by factors other than the treatment. 

#### 3.8.2. Clinical Indications, Dosage, and Outcome Variables

The method was used in horses to reduce stress. The study described the treatment in five different groups (*n* = 60; one control and four experimental groups) with music for one hour a day, music for three hours a day, massage on the day preceding a race, and daily massage during the six months of the racing season, plus a control group. The efficacy of the treatment was assessed by registering the horses’ heart rate (HR) and variables of heart rate variability (ratio of low to high frequencies of the power spectrum–LF/HF), and root mean square of successive beat-to-beat difference [RMSSD]), which were measured while preparing horses for training sessions. Salivary cortisol concentrations were measured before and after training sessions. Official general handicap and success coefficient in the racing season were considered as performance variables. 

#### 3.8.3. Clinical Effects

The study, with a high risk of bias, reports that playing relaxing music for three hours a day had more positive effects on horses’ emotional state than music for one hour, evaluated by heart rate (see Table 4).

### 3.9. Vibration Therapy 

Vibration therapy is when vibrations are transferred to the body through a contact surface that is in a mechanical vibrating state. 

Five RCTs involving 79 sound horses, three case studies involving 25 horses, and one case study involving 10 dogs were identified (see Table 5). 

#### 3.9.1. Study Quality

The risk of bias in the RCTs was judged as low to moderate (1) and moderate (4). The case studies were judged as having a moderate to high (2) and high risk of bias (2). 

#### 3.9.2. Clinical Indications

One case study concerned the treatment of horses with chronic lameness with whole-body vibration (WBV). One equine study looked at cycloidal vibration therapy on sound horses, while the remaining equine studies investigated the different types of physiological effects of WBV on sound horses. One case study examined physiological changes in sound dogs. 

#### 3.9.3. Interventions and Controls

In the case study on horses with chronic lameness, the horses stood on a vibration plate for 30 min a day, five days a week for 60 days, and there was no control group. The seven studies on sound horses described treatments varying from 10 min vibration (15–25 Hz) to 45 min, five days a week (30 Hz). The study on dogs used WBV exercise in daily sessions at 30 Hz for five minutes, followed by 50 Hz for five minutes, and finally 30 Hz in five minutes over five days. Acceleration range was 12–40 m/s^2^ and amplitude was 1.7–2.5 mm. The five RCT studies had control groups; the other studies reported changes before and after treatment.

#### 3.9.4. Outcome Variables

In the study on horses with chronic lameness, the horses were assessed by clinical examination and inertial measurement unit technique. The seven studies experimenting on sound horses used outcome tools such as radiographs measuring bone mineral content and laboratory blood analysis of different blood variables. The study on dogs used complete blood count and serum chemistry.

#### 3.9.5. Clinical Effect

In the study on horses with chronic lameness, there was no difference in lameness seen after 30 or 60 days of WBV (moderate-to-high risk of bias). The seven studies on sound horses (moderate to high risk of bias), which used radiographs measuring bone mineral content and laboratory blood analysis of different blood variables as outcome measures, showed a decrease in gamma-glutamyltransferase, serum cortisol, and creatinkinase values, but there were no changes in the other registered variables. The study on dogs, with high risk of bias, reported that the treatment did not cause adverse effects on hematology and serum biochemistry in healthy adult dogs (see Table 5).

## 4. Discussion

This systematic review demonstrates a large number of publications on a wide variety of CAVM therapies, accessible through database searches. However, the absolute majority of these publications are methodology or overview articles. Thus, for most therapies, there are few or no scientific articles describing the clinical efficacy of a single type of indication in the species cat, dog, or horse. Of the publications that met the inclusion criteria, the majority did not have scientific documentation of sufficient quality to draw conclusions regarding their effect, assessed as the studies’ ”risk of bias”. 

It is important that animal health personnel have a general knowledge of CAVM therapies in order to be able to answer questions from the increasing number of animal owners interested in CAVM. The respondents in a US survey of colleges and schools of veterinary medicine [54] stated that the level of public interest makes it necessary for veterinary students to be prepared for questions about CAVM, but that the inclusion of CAVM courses in veterinary medicine must be based on evidence. Regardless of the veterinarian´s private opinion on CAVM, knowledge about the scientific basis for information on the advantages and disadvantages of using CAVM is essential in situations where the animal owner addresses the matter. Otherwise, owners may choose to use CAVM without telling their veterinarian, as human patients tend to do with regards to their doctor [55]. Thus, there is a need for unbiased information regarding the scientific evidence for CAVM therapies.

In the present systematic literature review, 15 therapies out of 24 had no scientific documentation that matched the inclusion criteria, despite the criteria being broad. The reasons for the lack of scientific documentation can be speculated. For instance, there could be a lack of funding within the CAVM community, and limited experience of conducting research and the willingness to deposit means for research and development may be limited. Another explanation may be low demand for scientific evidence of efficacy, as few CAVM customers ask for scientific documentation. Further, there might be skepticism among CAVM therapists towards conventional medicine and the design of scientific studies. Further, an explanation for the lack of research may be that it is not possible due to the characteristics of a particular CAVM method. This applies especially for those therapies that have an explanatory model that is different from those used in conventional medicine; for example, the use of RCTs when studying individualized homeopathic treatment [7,31].

The results of the present review indicate that the included therapies are used in a variety of indications. Some therapies treat animals with a specific pathology, mainly skin disease, malignancies, or lameness. Others aim at affecting behavior or have defined physiological effects, such as enhancing bone mineral content. It is possible that indications for which conventional veterinary medicine offers an extensive program of treatment or recommends changes in training are more frequently treated with CAVM. As an example, back pain in horses is often regarded as difficult to treat [56], which may make veterinarians more likely to try CAVM as a complement to conventional medicine [5]. Thus, therapies such as acupuncture and chiropractic can be offered for back pain and if horse owners have an interest in CAVM [57]. Further, the range of services offered at veterinary clinics has historically been driven by demand from pet owners [6,57].

Despite the limited number of scientific publications, four studies were identified with a lower risk of bias, which indicates that the study result can be considered more reliable over time. Of these, no significant difference between treated and control group was reported in two studies each, on homeopathy and on gold implants, respectively. The two RCTs on gold implants into acupuncture points around hip joints in dogs with osteoarthritis showed no difference in the dogs’ movement. These negative findings are consistent with the results of a previous literature review including three RCTs, three retrospective, and five case studies [9], concluding that the case and retrospective studies, with a higher risk of bias, showed therapeutic success. However, the studies with a lower risk of bias showed no differences, and the single included RCT study with objective evaluation of the treatment effect showed no differences between treatment and placebo groups.

A similar pattern emerges from the studies on homeopathy The included studies, with moderate risk of bias, such as homeopathic hypotensive treatment in dogs with early, stage two heart failure and the study on cats with hyperthyroidism, showed no differences between treated and non-treated animals. An RCT on treatment of osteoarthritic dogs showed a difference in three of the six variables (veterinary-assessed mobility, two force plate variables, an owner-assessed chronic pain index, and pain and movement visually analogous scales). These results on homeopathy are supported by another systematic literature review of 18 RCTs, representing four species (including two dog studies) and 11 indications [7]. The authors excluded generalized conclusions about the effect of certain homeopathic remedies or the effect of individualized homeopathic intervention on a given medical condition in animals. In addition, the meta-analysis of nine homeopathy trials with a high risk of bias, and two studies with a lower risk of bias, conclude that there is very limited evidence that clinical intervention in animals using homeopathic remedies can be distinguished from similar placebo interventions [8].

Most studies on vibration therapy examined the physiological effect of treatment on sound horses. These studies had a moderate to high risk of bias and showed no effect on bone density, muscle activity or joint mobility, or stride length. The results are partly opposite to those from human medicine, where there are indications that vibration treatment results in increased bone density [58]. The scope and quality of the scientific documentation for other therapies was such that no conclusions can be drawn regarding the methods’ effectiveness in treating current animal species and indications.

As the aim of the literature review was to evaluate the efficacy of the treatments in sport and companion animals, the search was restricted to the species horses, dogs, and cats. It is likely that if more species had been included, additional publications could have been found. It is possible that some additional knowledge could have been extracted from these potential studies. However, extrapolating results between species can be unsafe, as different species have different pathologies and areas of use. Further, in order to ensure the specific efficacy of each method, no combination of therapies was included.

One major problem with drawing clinically applicable conclusions from the included articles is the diversity of study design. Less than half of the studies were RCTs, the minority used similar treatment protocols, and the main outcome variables were either subjective or measures that are easily influenced by other factors than the treatment, such as owner questionnaires, visual lameness examinations, or clinical examinations assessing heart or respiratory rates. Another problem is the small sample sizes with low statistical power and not allowing the detection of small differences between the treatment groups and controls. Therefore, most of the studies were classified as having moderate to high risk of bias based on templates developed by the Cochrane Collaboration [10] and SBU [11]. The grading criteria included study design, study population, type of control, confounding factors, classification/selection, deviation from planned therapy, lost to follow-up, type of outcome assessment, and relevance. It is unlikely that, with another type of grading, a different result would have been reached. Thus, it is most unlikely that the overall conclusion would have been different.

In a few of the CAVM studies, statistically significant results in favor of the active therapy were reported. In the evaluation of these results, it is essential to take not only statistical but also clinical significance (effect size) into account. When statistical differences were reported, either the effect size was of questionable clinical significance, or the results were not replicated in independent studies. Further, the description of the majority of studies did not enable reports of the minimal clinically important difference, defined as the smallest difference in score in any domain or outcome of interest that is perceived as beneficial.

### Limitations

The primary limitation for drawing conclusions on treatment efficacy from this systematic review is the lack of publications and high heterogeneity in terms of indications, applied techniques, treatment protocols, and outcome variables across studies.

Professional librarians performed the literature search, and the search words were selected after a pilot literature search. The broad search strategy that was used had low specificity; only 42 of 982 articles (4%) fulfilled the inclusion criteria. It is possible that a few, more marginal, search terms (for instance hirudotherapy added to leeches or clay to mud therapy) would have resulted in a limited number of additional articles to be reviewed in full text. It is, however, unlikely that more than sporadic such articles would have fulfilled the inclusion criteria at a final review.

A systematic literature review attempts to gather available empirical research by clearly defined, systematic methods to obtain answers to specific questions. Since the number of studies was low for each species/indication/therapy combination, a pooled statistical analysis with meta-analysis was not feasible. In addition, when a narrative approach is used, as we have done, it is difficult to obtain an interpretable overview of the scientific documentation due to the high heterogeneity of study designs.

## 5. Conclusions

The present systematic review has revealed significant gaps in scientific knowledge regarding the effects of a number of “miscellaneous” CAVM methods used in cats, dogs, and horses. For the majority of the therapies, no relevant scientific articles were retrieved. For nine therapies, some research documentation was available. However, due to small sample sizes, a lack of control groups, and other methodological limitations, few articles with a low risk of bias were identified. Where beneficial results were reported, they were not replicated in other independent studies. Many of the articles were in the lower levels of the evidence pyramid, emphasising the need for more high-quality research using precise methodologies to evaluate the potential therapeutic effects of these therapies. Of the publications that met the inclusion criteria, the majority did not have any scientific documentation of sufficient quality to draw any conclusion regarding their effect. Several of our observations may be translated into lessons on how to improve the scientific support for CAVM therapies. Crucial efforts include (a) a focus on the evaluation of therapies with an explanatory model for a mechanism of action accepted by the scientific community at large, (b) the use of appropriate control animals and treatments, preferably in randomized controlled trials, (c) high-quality observational studies with emphasis on control for confounding factors, (d) sufficient statistical power; to achieve this, large-scale multicenter trials may be needed, (e) blinded evaluations, and (f) replication studies of therapies that have shown promising results in single studies.

## Figures and Tables

**Figure 1 animals-11-03356-f001:**
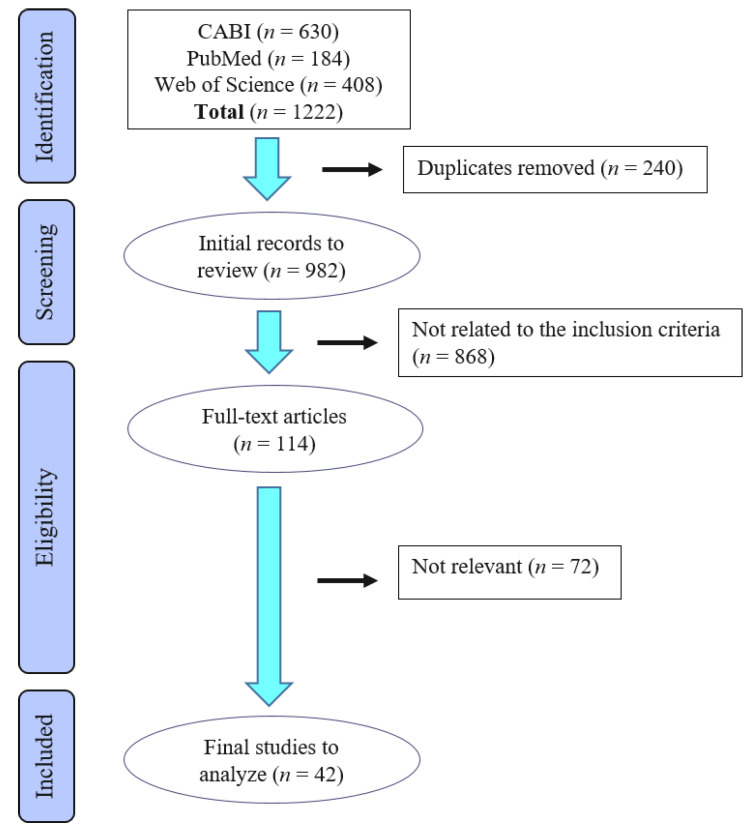
Flow diagram of the stages of the selection process used for identification of studies eligible for final analysis.

**Table 1 animals-11-03356-t001:** Aromatherapy.

Study	Study Design	Control Group	Study Sample	Intervention and Dosage	Outcome Variables	Main Results	Study Risk of Bias
Baldwin and Chea, 2018 [12]	RCT (exper-imental)	Cross over	9 horses Inclusion: Sound horses Exclusion: -	A. Humidified lavender oil and humidified air (control) B. Humidified chamomile oil and humidified air (control) 1 week washout.	Heart rate variability (HRV) Root mean square of successive differences (RMSSD) between the interbeat intervals (indicator of parasympathetic tone)	Lavender transiently increased RMSSD and reduced percentage of very low-frequency HRV oscillations immediately after treatment. Chamomile had variable effects, none of which reached significance.	High
Ferguson, Kleinman. Browning, 2013 [13]	RCT (experi-mental)	Cross over	7 horses Inclusion: Sound horses Exclusion: -	Aromatherapy-treated horses = humidified air with a 20% mixture of 100% pure lavender essential oil for 15 min. Control = humidified air.	Heart rate (HR) Respiratory rate (RR)	Change in HR after treatment was significantly greater after aromatherapy compared with the control treatment. The RR did not differ.	Moderate/high
Heitman, Rabquer, Heitman, Streu, Anderson, 2017 [14]	RCT (experi-mental)	Cross over	8 horses Inclusion: Sound horses Exclusion: -	During a trailer ride (stressor), the horses received humidified air as the control and lavender aromatherapy (LA) as the treatment.	Heart rate (HR) Cortisol Norepinephrine	The average difference between the baseline and stressed measurements of HR and cortisol increased in both groups when the horses were transported. There was no difference in the HRs of the control and treatment horses; there was a difference in cortisol levels, with lower levels in the treated group.	Moderate/high
Komiya et al., 2009 [15]	RCT (experimental)	Cross-over	5 dogs Inclusion: sound dogs Exclusion: -	Lavender oil (0.18 mL) or saline (0.9% NaCl) solution (0.18 mL) was topically applied to the inner pinnas of both ears of all dogs at 8:30, 12:00, 15:30, and 19:00 on day 2. Each trial was duplicated in each dog, with an interval of 3 to 4 days between trials.	An ambulatory electrocardiogram (ECG) monitor was placed on each dog and 48-h ECGs were recorded. Spectral indices of heart rate variability, power in the high-frequency range, and the ratio of low-frequency to high-frequency power were calculated as an indirect estimate of autonomic nerve activity.	When dogs were treated with lavender oil, the mean heart rate was significantly lower during the period of 19:00 to 22:30 on day 2 compared with when dogs were treated with saline solution. High-frequency power during the period of 15:30 to 19:00 was significantly higher when dogs were treated with lavender oil compared with when dogs were treated with saline solution.	Moderate/high

**Table 2 animals-11-03356-t002:** Gold therapy.

Study	Study Design	Control Group	Study Sample	Intervention and Dosage	Outcome Variables	Main Results	Study Risk of Bias
Abdoon et al., 2016 [16]	Case study	No	10 dogs and 6 cats Inclusion: spontaneous mammary tumor. Exclusion: -	Plasmonic photothermal therapy to evaluate the cytotoxic effect of intratumoral injection of 75 μg gold nanorods/kg of body weight followed by direct exposure to 2 W/cm^2^ near infra-red laser light for 10 min on ablation of mammary tumor.	Case history, clinical, ultrasound, and histopathological examination.	Results showed that 62.5%, 25%, and 12.5% of treated animals showed complete remission, partial remission, and no response, respectively. Tumor was relapsed in four cases of initially responding animals (25%). Overall survival rate was extended to 315.5 ± 20.5 days.	Moderate/high
Ali et al., 2015 [17]	Case study	No	5 dogs and 2 cats Inclusion: natural mammary gland tumors Exclusion: -	A regime of three low plasmonic photothermal therapy doses at two-week intervals that ablated tumors.	Histopathology, X-ray, blood profiles, and comprehensive examinations were used before and after treatment.	Histopathology results showed an obvious reduction in the cancer grade shortly after the first treatment and a complete regression after the third treatment.	Moderate/high
Goiz-Marquez et al., 2008 [18]	Case study	No	15 dogs Inclusion: epileptic seizures Exclusion: -	Gold wire implants in acupuncture points.	Clinically and with electroencephalographic (EEG) recordings, the number of seizures and seizure severity were compared before and after treatment.	There were no significant statistical differences before and after treatment in relative power or in intrahemispheric coherence in the EEG recording. However, there was a significant mean difference in seizure frequency and severity between control and treatment periods.	High
Hielm-Björkman et al., 2001 [19]	RCT	Yes	38 dogs Inclusion: osteoarthritis induced by hip dysplasia	Gold wire implants at acupuncture points around the hip joints. Control: three “holes” in the skin. Both groups: meloxicam when needed.	Dogs’ locomotion, hip function and signs of pain, radiographs. Registration of meloxicam medication frequency.	No differences between the treated and control groups.	Low/moderate
Jæger et al., 2006 [20]	RCT	Yes	78 dogs Inclusion: osteoarthritis induced by hip dysplasia Exclusion: No previous treatment of acupuncture	The gold implantation group had small pieces of 24-carat gold inserted through needles at five different acupuncture points, and the placebo group had the skin penetrated at five non-acupuncture points.	The owners assessed the overall effect of the treatments by answering a questionnaire, and the same veterinarian examined each dog and evaluated its degree of lameness by examining videotaped footage of it walking and trotting.	There were significantly greater improvements in mobility and greater reductions in signs of pain in the dogs treated with gold implantation than in the placebo group.	Moderate
Bolliger et al., 2002 [21]	RCT	Yes	19 dogs Inclusion: hip dysplasia Exclusion: -	Gold bead implantation and superficial needle punctures (control).	Gait analysis, kinetic and kinematic analysis. Questionnaire.	No differences in kinetic and kinematic variables were seen before or one and three months after.	Low/moderate
Jæger et al., 2007 [22]	RCT (non blinded)	Yes, 7 dogs served as controls	73 dogs Inclusion: osteoarthritis caused by hip dysplasia Exclusion: No previous treatment with acupuncture	In the long-term two-year follow-up study (from Jaeger et al., 2006), 66 of the 73 dogs had gold implantation and seven dogscontinued as a control group. The 32 dogs in the original placebo group had gold beads implanted and were followed for a further 18 months.	The owners assessed the overall effect of the treatments by answering a questionnaire, and the same veterinarian examined each dog and evaluated its degree of lameness by examining videotaped footage of it walking and trotting.	The pain-relieving effect of gold bead implantation observed in the blinded study continued throughout the two-year follow-up period.	High

**Table 3 animals-11-03356-t003:** Homeopathy.

Study	Study Design	Control Group	Study Sample	Intervention and Dosage	Outcome Variables	Main Results	Study Risk of Bias
Balbueno et al., 2020 [23]	RCT	Yes	30 dogs Inclusion: in stage B2 of myxomatous mitral valve disease Exclusion: pulmonary edema, undergoing heart treatment	Three groups of 10 dogs each. Hypotensive treatment with Crataegus oxyacantha at a potency of (1) 6cH and (2) Crataegus MT or (3) hydroalcoholic solution (placebo).	Echocardiogram, laboratory blood tests, systolic blood pressure. Follow up at 30, 60, 90, and 120 days after initiation of the therapy.	No differences between groups. Some significant differences in within- group analysis.	Low/moderate
Boehm, 2020 [24]	Case study	No	10 dogs Inclusion: Atopic dermatitis Exclusion: Skin infection	Combination product Adrisin, three times a day over three weeks.	Canine atopic dermatitis lesion index, pruritus visual analog scale.	No differences over the duration of treatment.	Moderate/high
Neumann et al., 2011 [25]	Prospective, observational open-label cohort study	Yes, active control	68 dogs Inclusion: osteoarthritic Exclusion: <1 year	Complex homeopathic preparation Zeel ad us.vet (one to three tablets orally per day depending on body weight) to carprofen (4 mg/kg body weight) over 56 days.	Symptomatic effectiveness, lameness, stiffness of movements, and pain on palpation were evaluated by treating veterinarians and owners.	Clinical signs OA improved significantly at all time points (days 1, 28, and 56) with both therapies.	High
Beceriklisoy et al., 2008 [26]	RCT	Yes, active control	38 dogs Inclusion: Clinically pseudo-pregnant bitches	Group I: Thuja occidentalis D30 (8 globules, three times a day, per os, *n* = 15); Group II: Urtica urens D6 (eight globules, three times a day, per os, *n* = 15); Group III: naloxone (control group, 0.01 mg/kg, twice daily, s.c., *n* = 8).	Animals were classified as no, mild, moderate, and severe according to the clinical signs of mammary glands and behavioral signs during the study. Bitches were examined at 3–5 days intervals by means of inspection and palpation until clinical signs resolved.	Concerning mammary gland scores, treatments yielded significantly higher success rates in Group I and Group II (100% in both groups) compared to the success rate observed in Group III (37.5%).	Moderate
Hielm Björkman et al., 2009 [27]	RCT	Yes, one active control and one placebo	44 dogs Inclusion: OA and hip dysplasia	Treatment: HPS Zeel ½-1 ampoule/day Active control: Carprofen 2 mg/kg twice daily Placebo: lactose capsule	Six variables: Veterinary-assessed mobility, two force plate variables, an owner-evaluated chronic pain index, and painand locomotion visual analogue scales (VASs).	When measured by dichotomous responses of ‘improved’ or ‘not improved,’ there were changes. Veterinary-assessed mobility, peak vertical force and pain VAS showed a significant difference in improved dogs per group between the treated group and the placebo group.	Low/moderate
Cracknell and Mills, 2008 [28]	RCT	Yes	75 dogs Inclusion: fear response to fireworks	A homeopathicremedy based on phosphorus, rhododendron, borax, theridion, and chamomilla (6C and 30C in 20% alcohol), and a ‘control’preparation of water and 20% alcohol.	Assessment of dog’s fear severity was based on the owner’s perception of their dog’s behavior before, during, and after they completed the trial period.	There were significant improvements in the owners’ rating of 14/15 behavioral signs of fear in the placebo treatment group and all 15 behavioral signs in the homeopathic treatment group.	Moderate
Aboutboule, 2009 [29]	Case study	No	15 cats Inclusion: Eosinophilic granuloma complex (EGC)	The snake remedies used were Lachesis (9 cases), Crotalus cascavella (1 case), Crotalus horridus (1), Cenchris contortrix (1), Elaps corallinus (1), Naja (1), and Vipera (1).	Diagnosis of EGC was based on the clinical observation of characteristic dermatological lesions and usually confirmed by biopsy.	10 had good response, four dropped out and one did not respond to treatment.	High
Chaudhury & Varshney, 2007 [30]	Prospective study	Yes, active control	33 dogs Inclusion: Canine babesiosis	Group A was treated with C. horridus 200C, four pills four times a day orally for 14 days and Group B with diminazene aceturate at 5 mg/kg intramuscularly single dose. All the dogs were administered 5% dextrose normal saline at 60 mL/kg intravenously for four days.	The therapeutic efficacy was evaluated using clinical score, peripheral blood smear examination, and hematological indices (Hb, PCV and TEC) on days 0, 3, 7, and 14.	Mean clinical score revealed that numbers of clinical signs reduced significantly on day 14 post therapy with both C. horridus and diminazene aceturate. The number of parasitized erythrocytes also reduced significantly after treated with C. horridus on day 14 and with diminazene aceturate on days 3, 7, and 14 from baseline.	High
Hill et al., 2009 [31]	Case study	No	20 dogs Inclusion: atopic dermatitis	Individualized remedies prescribed on the basis of the dog’s cutaneous signs and constitutional characteristics.	The response to treatment was assessed by scoring the severity of pruritus from 0 to 10 on a validated scale.	In 15 cases, the owners reported no improvement. In the other five cases, the owners reported the treatment as associated with reported reductions in pruritus scores ranging from 64 to 100%.	High
Yaramiş et al., 2016 [32]	Case study	No	16 horses Inclusion: stereotypic behavior	Homeopathic Ignatia and/or Gelsenium have been used for every horse according to the effects of each remedy on behavioral problem. Additionally, Stramonium, Phosphorus, Nux vomica, Pulsatilla, Hypericum, Lycopodium, Argentum nitricum, Staphysagria, Arsenicum album, Lachesis, and Thuya occidentalis were used as treatment remedies specific for each horse.	For each horse, each person on the observation team was asked to provide their impression of the pattern of stereotypic behavior at the end of each month, according to daily observations throughout the study.	By treatment survey analysis, after one-month evaluation, the symptoms of stereotypical behaviors were found to be decreased, and after two months considerable regression was detected.	High
Raj et al., 2020 [33]	RCT	Yes	16 dogs Inclusion: oral papillomatosis	Homeopathic drugs in combination (Sulfur 30C, Thuja 30C, Graphites 30C, and Psorinum 30C) and placebo drug (distilled water) was administered orally twice daily for 15 days.	Dogs were clinically scored for oral lesions on days 0, 5, 7, 10, 15, 20, 25, 30, 45, 60, 90, 120, and 150 after initiation of treatment. Physical examination, complete blood count, and serum biochemistry. Biopsy samples from papillomatous lesions on 0th and 7th days post-treatment.	The homeopathic treatment group showed early recovery with a significant reduction in oral lesions reflected by clinical score in comparison to placebo-treated group.	Moderate
Bodey et al., 2017 [34]	RCT	Yes	40 cats Inclusion: Feline hyper-thyroidism, thyroid hormone T4	Individual remedies by adding the sarcode thyroidinum and an appropriate individualized simillimum. The placebo was water and ethanol.	After 21 days, the T4 levels, weight (Wt), and heart rate (HR) were compared with pre-treatment values.	There were no differences in the changes seen between the two treatment arms following placebo or homeopathic treatment, or between means of each variable before and after placebo or homeopathic treatment.	Low
Marchiori et al., 2019 [35]	RCT	Yes	10 dogs Inclusion: sound dogs, treated to stimulate canine immunity by modulating blood cell responses	The treated group (*n* = 5) received a basal diet with an additional dose of 0.5 mL/animal/day homeopathic solution (Echinacea angustifolia 6 CH, Aconitum napellus 30 CH, Veratrum album 30 CH, Pyrogenium 200 CH, Calcarea carbonica 30 CH, and Ignatia amara 30 CH), and Group C (*n* = 5) received only the basal diet.	The animals were weighed, and blood samples were collected for complete blood counts and serum biochemistry on days 1, 15, 30, and 45 of the experiment.	Lymphocyte counts were greater in the treated group on days 30 and 45 of the experiment.	High
Faulstich 2006 [36]	Case study	Yes, active control	46 Horses Inclusion: lameness Exclusion: -	Control: Hyaluronic acid Treatment: Complex of 14 homeopathically-prepared ingredients Zeel (D8).	Clinical examination days 7, 14, and 21. At two different horse clinics.	Therapeutic effect ascertained by clinical examination.	Moderate/high
Cayado Robledo, 2016 [37]	Case study	No	5 horses Inclusion: Equine laminitis	Homeopathic treatment: Aconitum 30C, Apis 15C, Arnica 7C, Belladonna 9C, Bryonia 9C, and Nux vomica 9C. Two granules of each component every hour during the day, 10 times per day for 10 days.	Variables evaluated included signs of pain, grade of lameness, digital pulse, and plasma levels of nitric oxide, nitric oxide synthase expression, carbon monoxide, and heme oxygenases.	Homeopathically-treated horses showed an obvious improvement after one day of treatment.	High

**Table 4 animals-11-03356-t004:** Leeches, Mesotherapy, Mud therapy, Neural therapy, and Music/Sound therapy.

Leeches (Hirudotherapy)							
Study	Study Design	Control Group	Study Sample	Intervention and Dosage	Outcome Variables	Main Results	Study Risk of Bias
Rasch, 2010 [38]	Cohort, Retrospective	No	57 horses Inclusion: Laminitis Exclusion: NA	112 bloodsucker applications in 57 laminitic horses.	Grading according to Obel.	Clinical improvement: 84% improved after the application of hirudotherapy.	High
Mesotherapy							
Alves et al., 2018 [39]	RCT (experi-mental)	Yes, active control	15 dogs Inclusion: chronic back pain	Control (CG; *n* = 5) and treatment groups (TG; *n* = 10). A combination of 140 mg lidocaine, 15 mg dexamethasone, and 20 mg thiocolchicoside was administered to group TG along with a 70-day course of a placebo. Carprofen was administered to Group CG for 70 days. On day 0, an intradermal injection of Ringer’s lactate was administered.	Canine Brief Pain Inventory (CBPI) and the Hudson Visual Analogue Scale (HVAS), evaluated before treatment (T0), after 15 days (T1), and at one (T2), two (T3), three (T4), four (T5), and five (T6) months.	No differences were found in CBPI results between groups TG and CG at T0 through T3 and in T6 and T7. Differences in CBPI sections after the discontinuation of carprofen: at T4 for Pain Interference Score for Pain Severity Score and T5 for PIS and for PSS, with group TG having overall better results. Individual treatment results were considered successful in one dog of group CG, whereas in group TG success was higher. No differences were registered with the HVAS.	Moderate
Alves et al., 2021 [40]	Retro-spective study	Yes	20 dogs Inclusion: back pain	1. combination of lidocaine, dexamethasone, and tiocolchicoseide. 2. as previous with an additional traumeel LT.	The Canine Brief Pain Inventory (CBPI) and the Hudson Visual Analogue Scale (HVAS), evaluated before treatment (T0), after 15 days (T1), and at one (T2), two (T3), three (T4), four (T5), and five (T6) months.	No differences were observed between groups.	Moderate/high
Mud therapy							
Bartos et al., 2014 [41]	Case study	No	10 horses Inclusion: sound horses	Horses were treated with mud treatment from Lake Hévíz 10 times, twice daily.	Before and after the experiment and eight weeks following, the average stride length and the longest distance between the hind and front foot during walking and trotting, and maximal flexibility of knee, hock and fetlock joints were measured. The maximal flexibility of each joint was measured with a joint protractor.	The stride length and longest distance between front and hindlimb were slightly but positively influenced after treatment.	High
Neural therapy (NT)							
Bravo-Monsalvo et al., 2008 [42]	Case study	No	18 dogs Inclusion: canine atopic dermatitis	One set given by injecting an intravenous dose of 0.1 mg/kg of a 0.7% procaine solution, followed by 10 to 25 intradermal injections of the same solution in a volume of 0.1–0.3 mL per site. Dogs were given six to 13 sets.	The dermatological condition of each patient was evaluated before and after the treatment using two scales: the pruritus visual analogue scale (PVAS) and the canine atopic dermatitis extent and severity index (CADESI).	The reduction of pruritus was statistically significant.	High
Eisenmenger et al., 1989 [43]	Case study	No	60 horses Inclusion: pain syndrome in the loin and hip region.	5 mL of a 1% solution without additives for each point; usually eight to 14 segments were infiltrated symmetrically paramedian. This infiltration was repeated each third day, four to five times.	Clinical examination	Of the 60 patients, 51 were infiltrated, of which 45 were controlled. Seven of them were no longer used for competition, and in four horses, the evaluation time after treatment was too short. Of the remaining 34 horses, 26 could be trained successfully and won several more races, while eight horses did not recover.	High
Sound (music) therapy							
Kedzierski et al., 2017 [44]	RCT	Yes	12 horses out of 60 Inclusion: Sound race horses Exclusion: -	Sixty horses were equally divided into one control group and four experimental groups; treated with music for one hour a day, music for three hours a day, massage on theday preceding a race, and daily massage during the six months of the racing season.	Heart rate (HR) and variables of heart rate variability, root mean square of successive beat-to-beat difference [RMSSD]) were measured. Salivary cortisol concentrations were measured before and after training sessions. Official general handicap and success coefficient in the racing season were considered as performance variables.	In the experimental groups, lowered HR, LF/HF, and salivary cortisol concentrations, as well as increased RMSSD, were found at various levels. It was shown that playing relaxing music for three hours a day had more positive effects on horses’ emotional state than for one hour.	High

**Table 5 animals-11-03356-t005:** Vibration therapy.

Study	Study Design	Control Group	Study Sample	Intervention and Dosage	Outcome Variables	Main Results	Study Risk of Bias
Buchner et al., 2017 [45]	Case study	No	10 horses Inclusion: sound horses	Standing on a vibration plate = control. Treatment = 15 and 25 Hz during 10 min, respectively.	Frequency, peak-to-peak displacement, and peak acceleration. Activity of m triceps, quadriceps, and longissimus dorsi was assessed with surface electromyography. Maximal body temperature at upper forelimb, thigh, and back was measured.	The 10-min vibration exercise had no significant effect.	Moderate/high
Halsberghe, 2017 [46]	Experi-mental single subject repeated design	4 horses	8 horses Inclusion: chronic lameness	8 horses were subjected to whole-body vibration (WBV) 30 min, twice daily, for five days a week for 60 days.	Visual examination and inertial sensors (lameness locator).	No significant difference in lameness was seen after 30 or 60 days of WBV.	Moderate/high
Mackechnie-Guire et al., 2018 [47]	RCT	Yes	30 horses Inclusion: sound horses	Treatment = 20 min cycloidal vibration therapy. Control = no treatment.	Inertial sensors, epaxial muscle dimension by a flexible curved ruler.	Within groups: there was a significant increase in muscle dimension and in inertial measurement unit registrations. No comparisons were made between groups.	Moderate
Carstanjen et al., 2013 [48]	Case study	No	7 horses Inclusion: sound horses	WBV treatment for 10 min and 15–21 HX.	Clinical variables and venous blood samples before and after treatment.	Decrease in serum cortisol and creatin-kinase values.	High
Santos et al., 2017 [49]	Case study	No	10 dogs Inclusion: sound dogs	WBV exercise with daily sessions at 30 Hz for five minutes, followed by 50 Hz for five minutes, and finally 30 Hz for five minutes over five days. Velocity 12–40 m/s^2^ and amplitude 1.7–2.5 mm.	Complete blood count and serum biochemistry.	The treatment did not cause adverse effects on hematology and serum biochemistry in healthy adult dogs.	Moderate/high
Nowlin et al., 2018 [50]	RCT	Yes	6 horses Inclusion: sound	Treated horses stood on a platform vibrating at 50 Hz for 30 min, and control horses stood on an adjacent platform that was not turned on for 30 min.	Lameness score, joint range of motion, and stride length were assessed visually. Horses were re-evaluated acutely after one initial 30-min treatment and again after three weeks, with treatments repeated daily (five days per week).	Findings suggest no differences from pre- to post-treatment between vibration therapy (VT) and control (CO) groups in any variables measured.	Moderate
Hyatt et al., 2017 [51]	RCT	Yes	20 horses Inclusion: sound horses	Treatment on a vibration plate at 50 Hz for 30 min, five days a week. Control = 30 min turnout.	Serum blood analysis	Gamma-glutamyltransferase showed a greater reduction in the control group compared to the treated group. Creatinkinase showed a reduced value in the control group and increased value in the treated group.	Low/moderate
Maher et al., 2017 [52]	RCT	Yes	11 horses Inclusion: sound horses	Treatment at 50 Hz for 45 min, five days a week. Both groups = exercise on a mechanical panel exerciser.	Radiographs were taken at −28, 0, and 28 days to assess bone mineral content. Heart rate and stride length at day 23.	No significant differences were found.	Moderate
Hulak et al., 2015 [53]	RCT	Yes	12 horses Inclusion: sound horses	Treatment at 50 Hz for 45 min, five days a week. Controls = exercise on a mechanical panel exerciser.	Radiographs were taken at −28, 0, and 28 days to assess bone mineral content.	No significant differences were found.	Moderate

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
