# Peer review of "A Systematic Review of Complementary and Alternative Veterinary Medicine: “Miscellaneous Therapies”"

_animals, 2021, doi:10.3390/ani11123356_

Round 1

Reviewer 1 Report

With interest, I have been reading the manuscript entitled: "A Systematic Review of Complementary and Alternative Veterinary Medicine: “Miscellaneous Therapies " by Spinella et al.

Congratulations on the manuscript. I have imagined how much hard work was involved, and I suppose it may have contributed to other researchers' training within the research group. Overall, the systematic literature review will contribute to complementary and alternative veterinary medicine (CAVM) knowledge and is worthy of publication. The researchers could present more details on how they analyze the biases/risk in low, moderate, and high.

Suggestions for improving the manuscript:

Simple Summary/Abstract

My comments: These two topics are very similar in themselves. Sometimes I know how hard it is to write them distinctly.  Please clarify it and elaborate a bit more here, trying to change the "Simple Summary". As a suggestion, I would remove "Relevant articles were assessed for scientific quality, and information was extracted on study characteristics, species, type of treatment, indication and treatment effects. Of 982 unique publications screened, 42 were eligible for inclusion, representing nine different CAVM 19 therapies: aromatherapy, gold therapy, homeopathy, leeches (hirudotherapy), mesotherapy, mud, neural therapy, sound (music) therapy and vibration therapy" information and add more specific results on cats, dogs, and horses.

Mu comments: Try starting the "simple summary" like this: "The results of the present review indicate that the included miscellaneous therapies are used in a variety of indications"...

Abstract:

My comments: Overall, I feel that the review question is not addressed adequately in the abstract onset.

I would like to see a brief rationale for why the complementary and alternative veterinary medicine (CAVM) evidence for clinical efficacy.  Also, the abstract does not contain any data or indication of bias/risk classification.

Introduction

My comments: Well written and straightforward.

Materials and Methods

My comments: Well written and straightforward. However, please clarify the assessment of the risk of bias. This information has been widespread throughout the manuscript. How did you go about classifying the evaluation of the risk of bias? More details are necessary.

Results and discussion

L 177-179: “For horses, the most commonly studied treatment was stress reduction with aromatherapy (3), followed by music therapy (?) different... (?)  types of lameness treated with vibration therapy (1) and leeches (1)”.

My comments: Please, revise this sentence and clarify it.

Sound (music) therapy

L468: “cofounding factors influencing the results”.

My comments: Please, provide more details about it.

L483: “ in this high risk of bias study”

Whole-Body Vibration

L 507: 12-40 m/s2 is velocity or acceleration?

My comments: To the best of the reviewer’s knowledge, only we can measure the vibrating platform amplitude, acceleration, and frequency. Please check it.

My comments: In addition, to establish the intervention protocol through mechanical vibrations, it is necessary to understand all variables, such as physical parameters (frequency, amplitude, and acceleration), exposure time, and position on the oscillating vibrating platform. Also, how does the platform move? The platform may move in different directions as vertical, bi-planar, or horizontal. To the best of the reviewer’s knowledge, only we can measure the vibrating platform amplitude, acceleration, and frequency. Please check it, and these aspects should be included when discussing and reviewing the current study results. See: doi.org/10.1177/2055668319827466

Author Response

A word document has been attached with the answers to reviewer 1´s comments.

Reviewer 2 Report

The authors used the Cochrane guideline for relevant literature,  professional searches in web of science core collection, CABI and PubMed.

The complementary and alternative veterinary medicine specialy homeopathy is very popular in dog, cat and horse owners but correct evidence based medicine is missing. Therefore bias is very high. This is discussed  very well in this paper!   Based on the treatment and controll groups of different alternative treatment possibilities this paper included only these literature examples.They included only these literature papers in which controll and treatment groups were available. 

This sentence of the conclusion is most important, " Of the publications that met the inclusion criteria, the majority did not have any scientific documentation of sufficient quality to draw any conclusion regarding their effect."These references include all different alternative treatment methodes which were described!I think the tables are very important because these represent the study design until the study risk of bias!

Author Response

Thank you for the encouraging words!  We have made the following additions to the text based on our interpretation of your comments.

Changes in the manuscript

Of the publications that met the inclusion criteria, the majority did not have any scientific documentation of sufficient quality to draw any conclusion regarding their effect.

Reviewer 3 Report

Firstly thank you to the authors for providing this SR. It is much needed and hopefully will provide a lot of advice to future researchers in this area.  This was not a small task but valuable information to be collated for those involved in animal therapies, both veterinary and non-veterinary professionals.

I do have a few comments:

Line 50: reference to evidence for the difference between explanatory models needed her. Also an example might be useful.
The second half of the first paragraph reads like the conclusion.
Line 70: you state that veterinarians should be well prepared for questions but that only a minority have studied cavm at college. Perhaps it should read something along the lines of - due to the demand from owners vets need to be prepared but are not likely to be.
Line 78/89: make this point clear that it relates to animals.
Line 85-88: how were these interventions selected? Mesotherapy is a more common veterinary intervention and I am not sure it falls into the same category. I would also argue that therapeutic touch is an soft tissue technique applied by hands (according to the definition by Haussler) perhaps you mean Light touch?
Line 120: did you exclude abstracts were there was not a full text article available?
Line 215: You have stated significant effects in this paragraph on clinical effects. I believe you are reporting on statistical significance and not clinical significance. Please can you make this clear.
Line 247: diodlaser should be diode laser
Page 8, table 2. Hielm-bjorkman paper: I believe it is a spelling mistake for Meloxicam - not melaxicam.
Line 270: space needed after Homeopathy and I question if it is a medical system!
Line 372: remove the underscore from the end of the line.
Line 390: remove line space
Line 407: TG should be in parentheses
Line 432 and the subsequent section: Does neural therapy fit in your classification of CAVM? In line 446 can you explain further what the 1% solution contained and what the segments were?
Line 483: what were the effects of the massage provided to the racehorses?
Table 4: Barros et Al, main results section - remove the section full stop.
Table 5: Mackechnie-Guire et al 2018 studied a very different form of vibration therapy to the WBV and this difference should be noted in the respective intervention paragraph from line 502
Line 530: remove the word miscellaneous
Line 541: please expand on your assumption that the veterinarian may be negative towards CAVM
Line 549/50: restructure this sentence to avoid raising a question
Line 551: What evidence do you have for a lack of funding and limited experience in conducting research?
Line 561: remove the word miscellaneous
Line 583: explain what you mean by 'the same'
Line 622: It would be good to include discussion regarding the common feature of low participant numbers/sample sizes and how this affects bias.

I would value discussion regarding effect size (cohens-d/hedges-g) for studies that do have a significant difference. I also urge you to comment on minimal clinically important difference, which is a requirement in human therapeutic reports.
Please can you include a summary of recommendations/key points that can serve to encourage raising the quality of future studies.

Author Response

A word document with the answers to reviewer 3´s comments has been attached.

Round 2

Reviewer 3 Report

Thank you for revising your paper and providing answers to my questions. I look forward to your paper on massage.